# Spatiotemporal immune atlas of a clinical-grade gene-edited pig-to-human kidney xenotransplant

Matthew D. Cheung [1,7], Rebecca Asiimwe [1,7], Elise N. Erman[1], Christopher F. Fucile[2], Shanrun Liu[3,4], Chiao-Wang Sun[3,4], Vidya Sagar Hanumanthu[4], Harish C. Pal[4], Emma D. Wright [1], Gelare Ghajar-Rahimi [1], Daniel Epstein[1], Babak J. Orandi [1], Vineeta Kumar [5], Douglas J. Anderson [1], Morgan E. Greene [1], Markayla Bell[1], Stefani Yates[1], Kyle H. Moore [5], Jennifer LaFontaine[1], John T. Killian Jr. [1], Gavin Baker [1], Jackson Perry [1], Zayd Khan[1], Rhiannon Reed[1], Shawn C. Little [1], Alexander F. Rosenberg[2,6], James F. George [1], Jayme E. Locke[1] & Paige M. Porrett [1] ✉

Pig-to-human xenotransplantation is rapidly approaching the clinical arena; however, it is unclear which immunomodulatory regimens will effectively control human immune responses to pig xenografts. Here, we transplant a gene-edited pig kidney into a brain-dead human recipient on pharmacologic immunosuppression and study the human immune response to the xenograft using spatial transcriptomics and single-cell RNA sequencing. Human immune cells are uncommon in the porcine kidney cortex early after xeno-transplantation and consist of primarily myeloid cells. Both the porcine resident macrophages and human infiltrating macrophages express genes consistent with an alternatively activated, anti-inflammatory phenotype. No significant infiltration of human B or T cells into the porcine kidney xenograft is detectable. Altogether, these findings provide proof of concept that conventional pharmacologic immunosuppression may be able to restrict infiltration of human immune cells into the xenograft early after compatible pig-to-human kidney xenotransplantation.

Kidney allotransplantation is a life-saving therapy for people with end-stage kidney disease, but there are not enough human kidneys to meet the growing demand[1,2]. Xenotransplantation is a promising solution to increase organ supply, and enthusiasm around xenotransplantation has recently increased in the wake of several reports of successful pig-to-human xenotransplants[3–5]. These key studies have been made possible by advances in gene editing technology that have removed carbohydrate xenoantigens from porcine cells which otherwise provoke binding of preformed antibody in humans and result in hyperacute rejection[6]. While significant progress has recently been made to overcome a major immunologic barrier to successful xeno-transplantation in humans, the safest and most effective

[1]Department of Surgery, University of Alabama at Birmingham, Birmingham, AL, USA. [2]Informatics Institute, University of Alabama at Birmingham, Birmingham, AL, USA. [3]Department of Biochemistry and Molecular Genetics, University of Alabama at Birmingham, Birmingham, AL, USA. [4]Flow Cytometry & Single Cell Core Facility, University of Alabama at Birmingham, Birmingham, AL, USA. [5]Department of Medicine, University of Alabama at Birmingham, Birmingham, AL, USA. [6]Department of Microbiology, University of Alabama at Birmingham, Birmingham, AL, USA. [7]These authors contributed equally: Matthew D. Cheung, Rebecca Asiimwe. ✉e-mail: pmporrett@uabmc.edu

immunosuppression strategy to control other human immunologic responses to the porcine kidney remains unclear. Nevertheless, this question is important to address prior to embarking on first-in-human clinical trials.

Important insights into which immunosuppression regimens may be effective in xenotransplantation have previously come from pig-to-non-human primate studies[7,8], with several regimens resulting in long-term survival of select animals. Despite these encouraging results, however, only 8 out of 119 NHPs (6.7%) survived for 12 months or more, and the median survival time for all NHP recipients of alpha-Gal knock out pig kidney in the published literature is only 24 days[9]. These outcomes in NHP xenograft recipients thus contrast markedly with the 93% patient and graft survival at one year observed in deceased donor allotransplantation[10] and suggest that immunologic outcomes have yet to be sufficiently optimized in NHPs to apply these immunosuppression strategies directly to human recipients. Nevertheless, a key lesson from the NHP experience that may inform immunosuppression strategies for humans is that many NHPs met humane endpoints because of complications of intense immunosuppression and not xenograft rejection[8,9]. This is perhaps not surprising given that most if not all pig-to-NHP xenotransplants have been performed in incompatible recipients, as the genetic knockdown of carbohydrate antigens that promotes crossmatch compatibility between pig cells and human sera has not resulted in compatible crossmatches between pig cells and NHP sera[11,12]. Moreover, limited survival of incompatible NHP recipients has diminished enthusiasm for agents like tacrolimus that are highly effective in controlling T cell responses in compatible human recipients but may nonetheless be less effective in incompatible transplantation where graft damage is driven instead by antibody[7]. It is thus unclear whether intensive immunosuppression regimens used in incompatible NHP models of xenotransplantation will be necessary to control human immune responses in the setting of crossmatch-compatible pig-to-human xenotransplantation. Improving our understanding of human immune responses to crossmatch-compatible porcine xenografts is thus critical to advance the field, as crossmatch-compatible transplantation will likely result in safest outcomes for human xenotransplant recipients[13].

In this work, we use single-cell and spatial transcriptomic approaches to investigate the human immune response to a genetically modified pig kidney. Given that safe and effective immunosuppression regimens for compatible pig-to-human xenotransplant recipients are thus difficult to define in incompatible animal models, we develop a preclinical model of pig-to-human kidney xenotransplantation to study human immune responses in the porcine kidney after crossmatch-compatible xenotransplantation. Using this model system, we further test the hypothesis that conventional immunosuppression used in allotransplantation would prevent the infiltration and expansion of human immune cells in the xenograft. Using transcriptomics approaches that could discriminate porcine from human cells in serial xenograft biopsy samples, we find that human myeloid cells infiltrated the xenograft cortex in limited numbers three days after transplantation and express an anti-inflammatory gene signature. Meanwhile, human adaptive immune cells are virtually undetectable in the porcine xenograft. Altogether, our study suggests that conventional immunosuppression may effectively control early human immune responses to a porcine kidney xenograft and helps address critical knowledge gaps surrounding the safety and efficacy of immunosuppression regimens in human xenotransplantation.

## Results

### Transplantation of a 10-gene modified porcine kidney to a brain-dead human decedent to assess the human immune response

To assess human immune responses to a porcine kidney xenograft in vivo, we developed a preclinical model of crossmatch-compatible pig-to-human kidney xenotransplantation. As previously reported,

porcine kidney xenografts were procured from a domestic pig with 10 genetic modifications[14] and transplanted into a nephrectomized, crossmatch-compatible, brain-dead human recipient[3] (Supplementary Fig. 1a). Immunomodulation of the recipient was established through both genetic modification of the porcine kidney as well as conventional pharmacologic immunosuppression, which included induction with anti-thymocyte globulin, rituximab, and methylprednisolone, followed by triple maintenance therapy with tacrolimus, mycophenolate mofetil, and prednisone (Table 1, see Supplementary Fig. 1b for tacrolimus levels and ref. 3). In addition to knockdown of the growth hormone receptor and three carbohydrate xenoantigens (GGTA1, B4GALNT2, CMAH), the porcine kidneys expressed human transgenes (CD55, CD46, THBD, PROCR, CD47, HMOX1) intended to reduce inflammation and prevent thrombotic complications within the kidney. As a primary goal of this study was to evaluate human immune responses in the setting of an immunosuppression regimen with established safety and tolerability in human patients, the regimen selected for this experiment mirrors the regimen used in crossmatch-compatible kidney allotransplant recipients at our transplant center. The kidney xenografts made urine but did not clear creatinine[3]. Although H&E sections of the xenografts revealed acute tubular necrosis and thrombotic microangiopathy of unclear etiology, there was no evidence of acute cellular rejection or binding of IgM, IgG, or complement proteins[3]. The experiment was terminated approximately 74 h after transplant due to hemodynamic instability[3].

Sequential needle core biopsies of the gene-edited porcine kidneys were taken immediately prior to transplantation, in situ on postoperative days 1 and 3, and immediately prior to explant (day 3T) (Supplementary Fig. 1c). Biopsies were taken from the left (day 1 and 3T) and right (pre-transplant and day 3) porcine xenografts. Although acute cellular rejection is not frequently observed at early time points in human allografts in recipients receiving induction therapy, we performed biopsies of the xenografts early and frequently after xenotransplantation for two reasons: 1) the kinetics of graft infiltration by recipient immune cells in human allotransplantation have not been well defined, and 2) the increased immunogenicity of the xenograft might provoke rapid immune infiltration even in the setting of systemic immune depletion. Given the limited amount of tissue available from the needle core biopsies, the number of analyses that we could perform on the samples was quite constrained. Although flow cytometry and immunofluorescence microscopy use fluor-conjugated antibodies which can identify immune populations in human kidneys[15,16], we were concerned about the ability of these reagents to identify human from porcine cells, and this approach was further limited by a lack of commercial reagents targeting porcine immune antigens. Moreover, the assessment of protein markers allows for examination of only a small number of epitopes on immune cells and therefore does not provide detailed information about potential states

**Table 1 | The dosage of pharmacologic agents given from post operative day (POD) 0 through 3**

| Immunosuppressive Medication | POD 0 | POD 1 | POD 2 | POD 3 |
|---|---|---|---|---|
| Anti-Thymocyte Globulin (Rabbit) | 175 mg | 175 mg | 175 mg | – |
| Rituximab | 1800 mg | – | – | – |
| Tacrolimus | – | 1 mg AM | 1 mg AM | 2 mg AM |
| | 1 mg PM | 1 mg PM | 2 mg PM | – |
| Mycophenolate Mofetil | – | 1000 mg AM | 1000 mg AM | 1000 mg AM |
| | 2000 mg PM | 1000 mg PM | 1000 mg PM | – |
| Methylprednisolone | 500 mg | 250 mg | 125 mg | 90 mg |

of cellular activation and/or differentiation. In contrast, next-generation RNA sequencing approaches permit an unbiased approach which can simultaneously screen thousands of genes and be highly specific given the ability to assess sequence-level differences[17], thereby allowing for the detection and discrimination of pig and human cell types. We therefore opted to analyze xenograft biopsies using spatial transcriptomics and single-nuclear RNA-sequencing (snRNA-seq) approaches. As detection of immune populations can be limited in samples analyzed by snRNA-seq[18], we also performed single-cell RNA-seq (scRNA-seq) on CD45+ immune cells that were FACS-enriched from a sample of the right explanted xenograft representing cortex through medulla (Supplementary Fig. 1c). Finally, single-nuclear RNA-seq was performed on wedge biopsies of the porcine kidneys at explant to generate libraries composed of larger numbers of parenchymal cell types. Details on read depth, UMI number, gene number, and other quality metrics for each sequenced sample are provided in Supplementary Fig. 2.

## A merged porcine-human reference genome distinguishes pig vs. human immune cells in the porcine kidney xenograft

In order to distinguish human from porcine cells, we aligned all sequenced reads to a custom human-porcine hybrid reference genome and annotated clusters based on established marker genes[19]. As discrimination of porcine from human immune cells was a key goal of this study and our ability to perform protein-based validation studies was limited by tissue and reagent availability, we assessed the specificity of our mapping to the hybrid reference genome with three different approaches. First, we aligned independent control biopsies from human and pig kidneys to the hybrid reference genome and found that key immune genes mapped to the appropriate species reference (Supplementary Fig. 3). Although these analyses evaluated the performance of our pipeline on samples derived from a single species, we hypothesized that pipeline performance might differ for mixed species samples (i.e., the xenograft). We therefore evaluated the mapping specificity of individual reads sequenced from CD45+ immune cells sorted from the explanted xenograft. Cells were initially clustered according to both cell type and species as individual genes had species-specific annotation after alignment to the hybrid reference (Fig. 1a, b and Supplementary Dataset 1). Myeloid cells comprised most of the detectable immune cells in the porcine and human compartments. There was little to no detection of human B or T lymphocytes (Fig. 1c–e).

We found that 97.8% of 18,833,529 transcripts recovered from 6513 immune cells associated with a single species, and 98.8% of cells possessed >90% of transcripts from a single species (Fig. 2). These analyses further revealed that ~10% of reads in pig macrophages mapped to 17 human genes (0.27% of human genes), likely a consequence of high sequence homology (Supplementary Fig. 4). Finally, to assess the impact of homology on the species assignment of individual cells, we employed an alternative mapping strategy using custom modified reference genomes composed of more species-specific genes (see "Methods" & Supplementary Fig. 5). We found a high correlation in the species assignment of cellular barcodes between the two methods (Supplementary Fig. 6), suggesting that the presence of gene homology did not significantly impact the species assignment for most cell types when using the hybrid reference genome. Collectively, these analyses highlighted the benefit of using the entire transcriptome to make cell assignments in lieu of individual genes that may not be specific to species or cell type.

## Spatial transcriptomics reveals limited human immune cell infiltration over the duration of the xenotransplant

Having established a pipeline by which we could confidently distinguish pig from human immune cells types, we used cell2location[20] to deconvolute our spatial transcriptomic data from the xenograft

biopsies into pig and human cell types based on input of reference transcriptomes. Reference transcriptomes were derived from sorted CD45+ cells and parenchymal cells from the porcine xenograft core and wedge biopsies (Supplementary Fig. 7). The snRNAseq data obtained from the wedge biopsies were used to generate the reference transcriptomes due to the low number of nuclei isolated from the core biopsies. We restricted the input to cell2location to reference transcriptomes recovered from the xenografts given the unknown impact of immunosuppression and/or ischemic injury on immune transcriptomes (Supplementary Fig. 8). However, since human adaptive immune cells were not well represented among CD45+ immune cells sorted from the xenograft (Fig. 1) and were thus not present in our reference transcriptomes, we attempted to identify human T and B cells by expression of multiple individual T and B cell genes known to be relatively specific to these cell types (Fig. 3). There were no detectable human T cells (Fig. 3a, b) or B cells (Fig. 3c) in any of the xenograft biopsies sampled at any time point after transplantation.

While porcine immune cells were readily detected in pre- and post-transplant xenograft biopsies, human immune cell types were not appreciably detected until post-transplant day 3 and were far less abundant than porcine immune cells (Fig. 4). As would be expected in the setting of effective T and B cell depletion[3], all human immune cells detected in the xenograft biopsies derived from myeloid lineages (Figs. 3 and 4). We next used cell2location to identify the spatial co-occurrence of the immune cells with kidney parenchymal cells to understand their locations within the kidney biopsies. Similar to other resident immune cell studies in mice and humans[21–23], the porcine macrophages and T cells were found near epithelial cells encompassing most structures of the kidney both pre- and post-transplant. At post-transplant day 3, when most human immune cells were detectable, human neutrophils and monocytes were found to co-localize predominantly with porcine endothelial cells, whereas human macrophages were mostly localized with porcine stromal cells (Fig. 4c). Altogether, these experiments reveal limited infiltration of the renal cortex by human immune cells.

## Pig resident and human infiltrating macrophages express genes consistent with an anti-inflammatory phenotype

As macrophages represented the most prevalent immune cell type in our biopsy data and are known to impact allotransplant outcomes[24,25], we interrogated macrophage activation state by examining expression of classically activated M1 and alternatively activated M2 genes in pig and human macrophage subsets among CD45+ cells sorted from the explant (Supplementary Dataset 2)[26]. We found increased individual gene expression (Fig. 5a, b) and composite gene expression scores of M2 compared to M1 genes in both pig and human macrophages (Fig. 5c). Although differences in gene expression between the species precluded comprehensive formal comparison of donor- versus recipient-derived macrophages, we performed a limited comparison of pro- and anti-inflammatory cytokines and select genes of interest known to be important in macrophage activation and function[27] (Fig. 5d). Collectively, these data suggest that macrophages populating the kidney xenograft express a more alternatively activated, anti-inflammatory transcriptome, independent of species.

## Discussion

Xenotransplantation is a promising conceptual solution to the organ shortage but has been previously limited in its clinical application by significant immunologic barriers between donor and recipient species. However, recent studies suggest that hyperacute rejection due to preformed antibody in humans can be overcome by the humanization of porcine kidneys through specific gene edits, and achievement of this milestone has renewed hope that xenotransplantation may advance to clinical trials soon[3–5,13,24]. Nevertheless, key questions remain surrounding immunosuppression in human recipients before

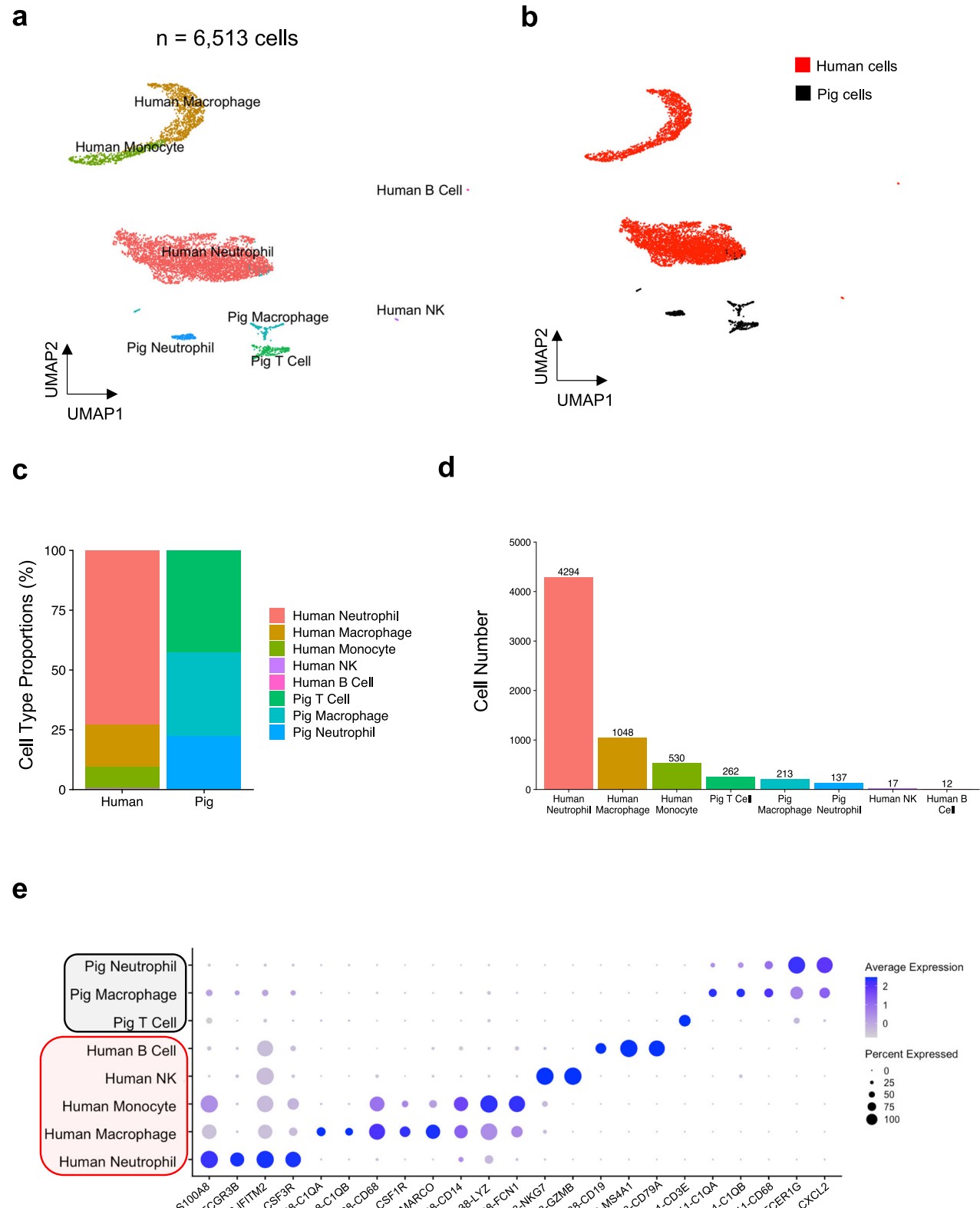

**Fig. 1 | Composition of CD45+ immune cells collected from the porcine xenograft explant.** The 10-GE porcine xenografts were removed from the brain-dead human recipient three days after xenotransplantation, and single-cell RNA-sequencing was performed on FACS-enriched immune cells from the right xenograft using pig- and human-specific CD45+ antibodies. Data were aligned to the hybrid human-porcine reference genome. a&b) UMAP of sorted CD45+ cells (*n* = 6513 cells) colored by cell type (**a**) and species (**b**). **c**, **d** Enumeration of sequenced human and porcine immune cells. **e** Expression of select marker genes in human and pig immune cell clusters.

**a**

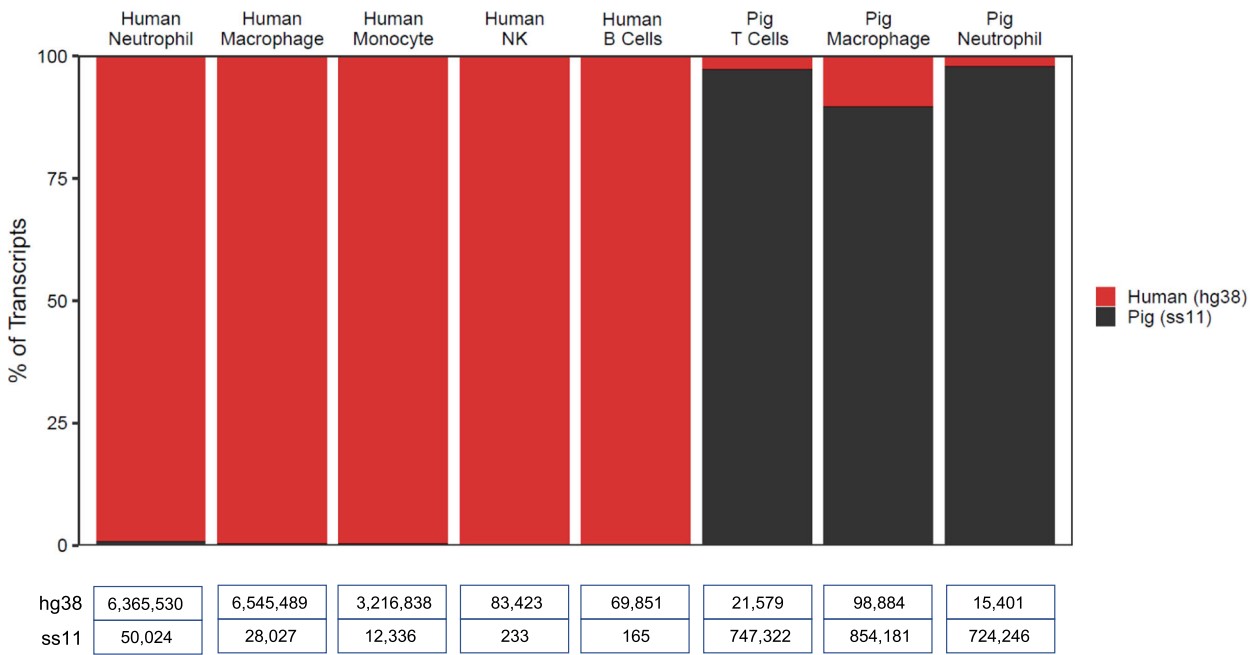

| | hg38 | ss11 |
|---|---|---|
| Human Neutrophil | 6,365,530 | 50,024 |
| Human Macrophage | 6,545,489 | 28,027 |
| Human Monocyte | 3,216,838 | 12,336 |
| Human NK | 83,423 | 233 |
| Human B Cells | 69,851 | 165 |
| Pig T Cells | 21,579 | 747,322 |
| Pig Macrophage | 98,884 | 854,181 |
| Pig Neutrophil | 15,401 | 724,246 |

**b**

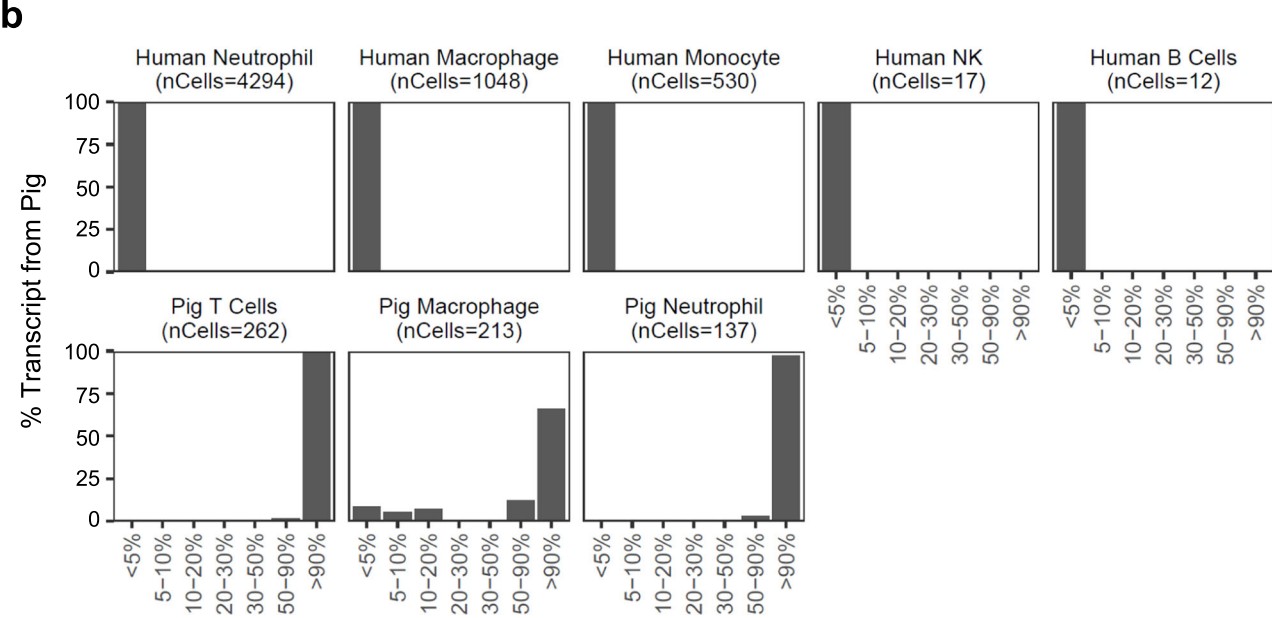

**Fig. 2 | Assessment of species mapping at the individual transcript and cellular levels.** Single-cell RNA-sequencing was performed on FACS-enriched immune cells from the xenograft explant as in Fig. 1. Data were aligned to the hybrid human-porcine reference genome. **a** Species origin of all transcripts for a given immune cell cluster. Table at the bottom enumerates the transcript number for each cluster that mapped to the hg38 (human) or ss11 (pig) portion of the hybrid reference genome. Additional details of pig macrophage transcripts mapping to the hg38 component of the porcine-human hybrid reference genome are given in Supplementary Fig. 4. **b** The frequency of cells that have low or high levels of transcripts derived from the pig.

first-in-human trials can be safely undertaken. Although prior work in NHP models of xenotransplantation has provided important insights into drug regimens which might attenuate immune responses to xenografts, the generalizability of these findings from NHP recipients to humans is limited by the inability to perform crossmatch compatible xenotransplantation in NHPs[11,28] which necessitates the use of intensive immunosuppressive regimens that are associated with high mortality rates[9]. There is thus a need for immune investigations in crossmatch-compatible recipients of porcine xenografts both to characterize immune responses to immunosuppressive agents in the setting of xenotransplantation as well as to determine tolerability in the human setting. Ideally, these studies would be performed in a model system where human immune responses to a gene-edited humanized kidney xenograft can be studied in vivo prior to the transplantation of living recipients. Herein, we addressed this knowledge gap by studying the human immune response to a gene-edited porcine kidney xenograft using a brain-dead human decedent model. Notably, we treated the human decedent xenograft recipient with a clinically utilized immunosuppression regimen to determine whether a regimen with a well-established safety and efficacy profile used in

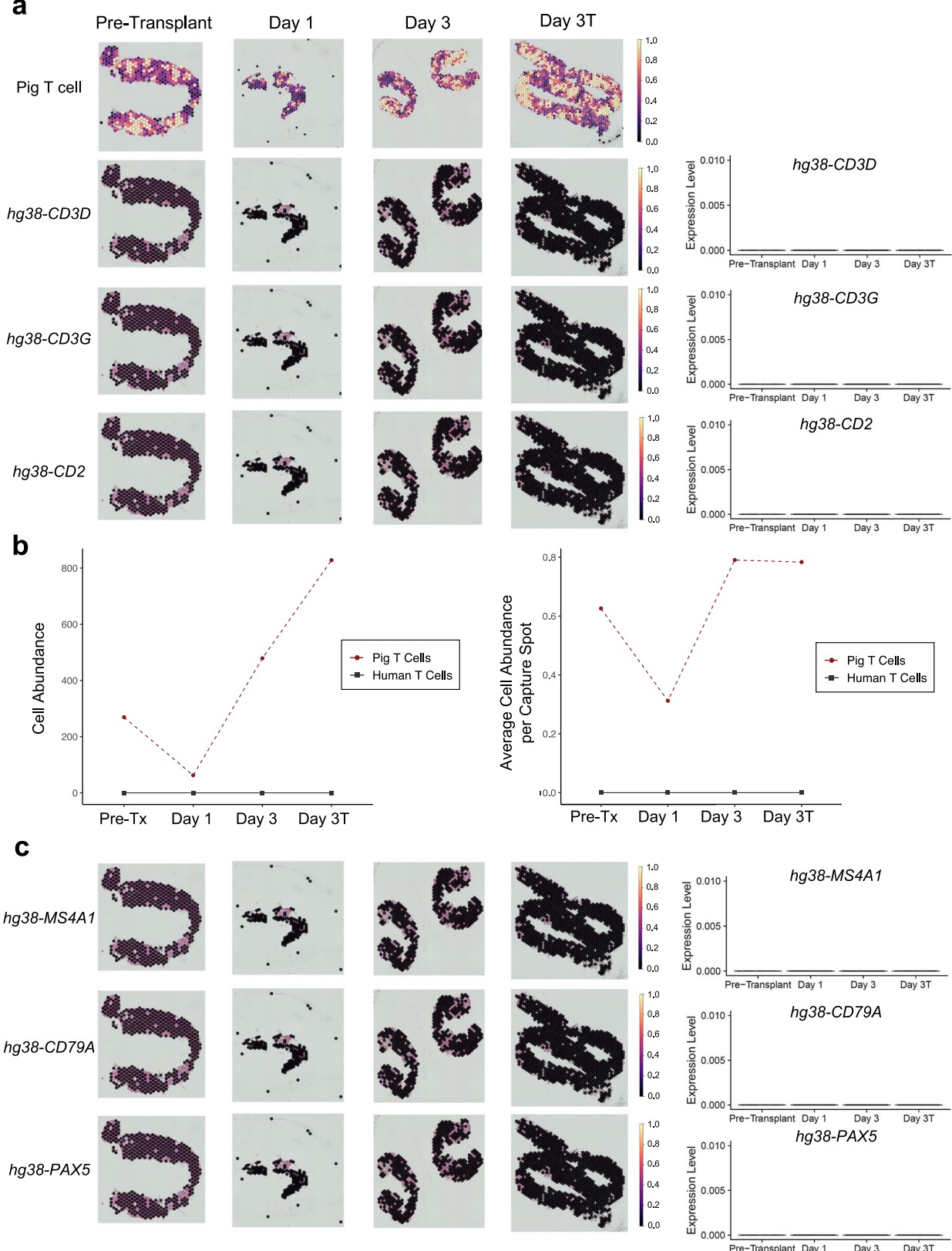

allotransplantation might be extended to the xenotransplantation setting.

Using single-cell, single-nuclear, and spatial transcriptomics approaches, we found that human immune cells were uncommon in the porcine kidney cortex early after xenotransplantation and consisted of predominantly myeloid cells. Human T cells and B cells were notably absent in the xenograft renal cortex at all assessed time points and were also not detectable among immune cells isolated from explant samples encompassing both the renal cortex and medulla. We attribute the absence of detectable human lymphoid cells in the kidney xenografts to effective systemic depletion of T cells and B cells by anti-thymocyte globulin and rituximab, respectively, although the short

**Fig. 3 | No detection of human T or B lymphocytes in the porcine kidney xenografts.** Spatial transcriptomics was performed on serial needle core biopsies of 10-GE porcine kidneys before and after transplantation into a brain-dead human recipient. Biopsies were obtained from either the right (pre-transplant and day 3 samples) or left (day 1 and day 3T) xenografts. Cell type signatures were identified from reference transcriptomes using cell2location (for Pig T cells, top plots in 3a) or expression of individual marker genes (e.g. *CD3E* and *CD19*) for adaptive immune cell types that were not well represented in CD45+ immune cells sorted from the xenograft explants (see Fig. 1) and were therefore not included in the reference transcriptomes passed to cell2location. **a** No detection of human T cell genes in xenograft biopsies at any time point (left) with quantification of gene expression levels (right). Pig T cells were readily detectable in the xenograft (top row). **b** Calculated total (*left*) and normalized (*right*) cell abundance for various pig T cells in the indicated biopsies. Note that cell abundance for human T cells was imputed from expression of *hg38-CD3E*, *ss11-CD19*, and *hg38-CD19* genes as shown (**a**). **c** No detection of human B cell genes in xenograft biopsies at any time point (left) with quantification of gene expression levels (right). Pre-tx pre-transplant. day 3T biopsy was taken on post-transplant day 3 at study termination.

duration of our experiment may not have provided the opportunity for significant human immune cell infiltration into the porcine xenografts. While the timeframe of our experiment limits our ability to draw firm conclusions about the impact of immunosuppression on infiltrating immune cells, our study nevertheless improves our understanding of early human immune responses to a porcine kidney xenograft and was necessary for the sole reason that observations from either animal models of xenotransplantation or human allotransplantation may not predict findings in this human xenotransplantation model. Moreover, the kinetics of human immune cell infiltration into a kidney early after allotransplantation have not been well studied in vivo given the difficulties with obtaining longitudinal biopsies in human allotransplantation as well as the inability to discriminate donor from recipient immune cells using conventional histologic techniques or immunofluorescence microscopy.

We found it reassuring that the potentially highly immunogenic or inflammatory microenvironment of the xenograft did not result in the rapid recruitment and accumulation of any human immune cell type into the renal cortex under the conditions tested in this experiment. Interestingly, however, we were able to capture a larger number of human myeloid cells by scRNAseq when the kidney medulla could be sampled after explant. This increase in the number of human myeloid cells captured from the renal medulla is not unexpected given reports that macrophages are more frequent in this compartment[22,23]. The signals responsible for recruitment of human myeloid cells into the porcine renal medulla are at present unknown but may be a consequence of ischemia-reperfusion injury. Many of the human immune cells co-localized with porcine endothelial cells and thus may represent blood contaminants or potentially a response to the endothelial damage noted in our prior study[3]. No matter the recruitment signals, it was additionally reassuring that both the pig and human macrophages predominantly expressed an anti-inflammatory gene signature, which may limit xenograft injury and support a rapid recovery of the porcine kidney after xenotransplantation. The anti-inflammatory macrophage gene signature could be explained by their proximity and potential interactions with stromal cells, which have been shown to skew macrophages towards this phenotype[29,30]. Additional studies performed over a longer time course in living humans will be needed to fully understand the implications of these data and determine to what extent these immune phenotypes are due to the administration of steroids versus the genetic modifications of the pig, such as insertion of the human *HMOX* gene[31].

Our ability to distinguish porcine donor from human recipient immune cell types was instrumental in characterizing the immune composition of the porcine kidney xenograft. To accomplish this task, we developed a porcine-human reference genome to properly align the RNA sequencing data and distinguish pig (donor) vs. human (recipient) immune cells within the transplanted porcine kidney. A similar strategy which leverages sequence-level differences in SNPs between donor and recipient has been utilized in allotransplantation to distinguish donor vs. recipient cells[17]. We also developed an alternative mapping strategy to validate our species and cell type assignments that accounts for sequence homology and overlapping reads (Supplementary Figs. 5 and 6). In doing so, we were able to assess the immune cell repertoire of a porcine kidney xenotransplant in a human in vivo at the sequence level; this approach further helped us avoid cell type and species assignment errors that might have occurred if we had relied instead on a limited number of protein epitopes. Additional comparative studies of the pig and human immunome in the future may allow us to develop a list of key marker genes with sufficient sequence dissimilarity that whole transcriptome approaches may no longer be necessary. Future improvements in porcine reagent development will also help us validate our findings at the protein level. In summary, we anticipate that the tools developed to distinguish porcine and human cells in this study will help inform cell type assignments and analyses of other transplantation experiments as well as other multi-species experiments in a broad range of investigations.

Despite these encouraging results, our study has several limitations. First, we were not able to determine the impact of immunomodulatory strategies at later time points due to the short duration of the experiment, which limits the interpretation of our data to early responses. Nonetheless, these findings remain important given that there are no other studies that have addressed the human immune response to a 10-gene edited porcine xenotransplant in vivo. Second, our data do not distinguish the specific impact of pharmacologic interventions from genetic edits within the pig kidney. Moreover, our study does not compare outcomes between immunosuppression regimens and thus cannot be used to predict which group of agents will result in best outcomes. Our study therefore does not preclude a role for other immunosuppression agents (e.g. co-stimulation blockade) in clinical xenotransplantation. Third, we may have overestimated infiltrating human cell number as we could neither distinguish cells in the kidney interstitium from potential blood contaminants nor correct for alignment error for homologous genes. We attempted to overcome these limitations by using the whole transcriptome to make species and cell type assignments to reduce the impact of homologous genes on our cell calls. We also focused our analyses on cells unlikely to represent blood contaminants (i.e., macrophages). Nevertheless, even if we have inaccurately identified 20% of pig macrophages as human cells (Supplementary Fig. 6), these data still suggest that infiltration of the xenograft by human immune cells known to promote rejection (i.e., T cells) or chronic injury (i.e., inflammatory macrophages) is limited. Fourth, given the small quantity of tissue and the lack of available porcine-specific reagents, we were unable to perform protein-level validation as discussed above. Advancements in the field will require improving the availability of anti-porcine antibodies and other species-specific reagents that can be used for validation studies. Finally, our study was performed in the milieu of brain death and must be interpreted cautiously in the larger context of functional outcomes in this model system. More specifically, we were unable to determine the impact, if any, of brain death physiology on our findings, and we expect that only future studies in living human recipients will be able to assess the potential impact of brain death physiology on the immune census of the porcine kidney. Moreover, the porcine xenograft did not clear creatinine in this experiment[3], and we observed histologic evidence of thrombotic microangiopathy (TMA) of unclear etiology on routine H&E within 24 h of transplantation[3]. It is therefore possible that the unknown suite of factors responsible for either

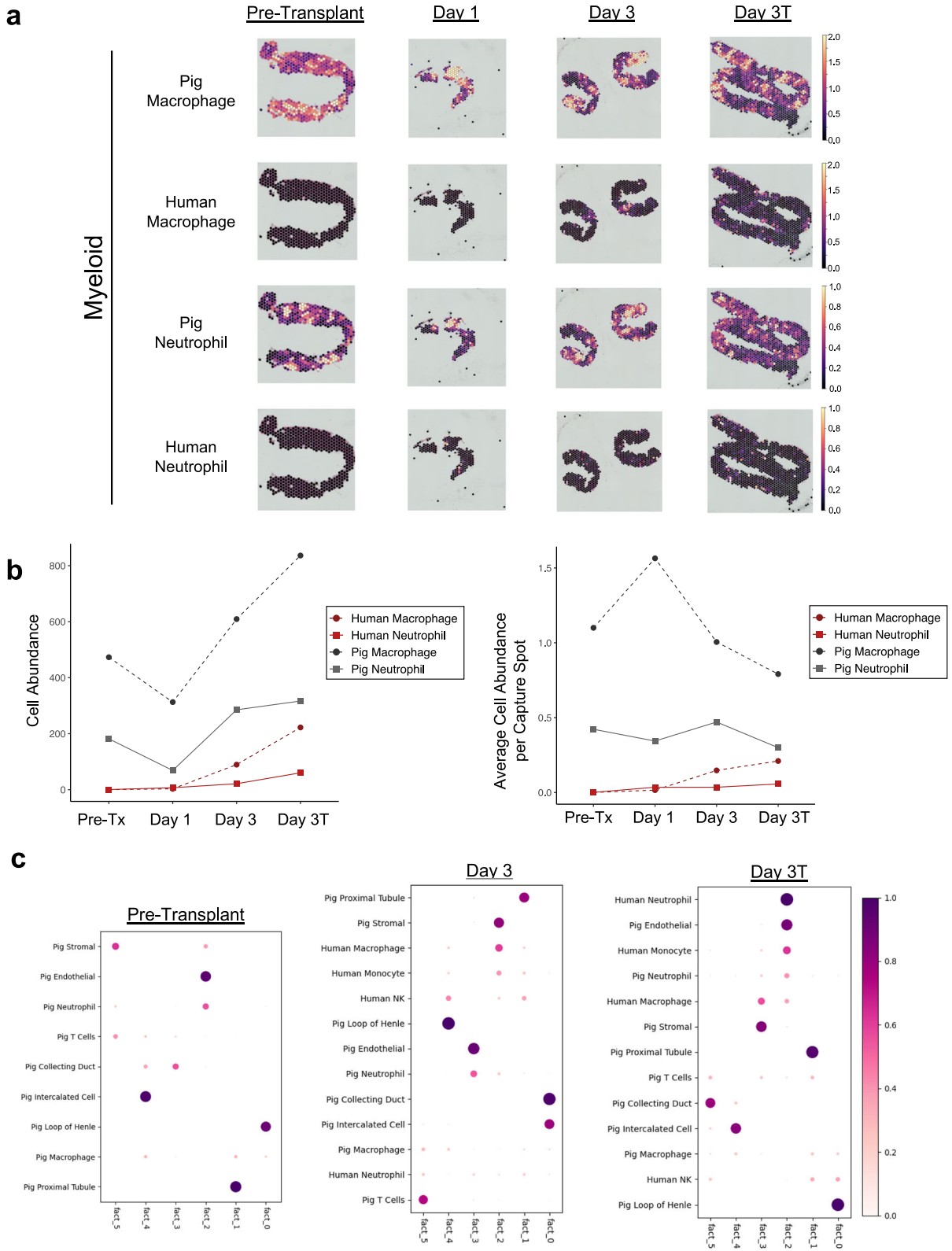

delayed graft function and/or TMA in this experiment may also have impacted our results. Additional immune investigations in either living recipients and/or brain-dead decedents with preserved xenograft function and normal histologic findings[32] will be necessary to fully address this issue.

In conclusion, we find limited infiltration of a gene-edited porcine kidney xenograft by human cells early after xenotransplantation in a brain-dead human decedent model treated with conventional immunosuppression. These observations suggest that the human adaptive and innate immune response to a porcine xenograft early after xenotransplantation may be controlled by currently available genetic and/or conventional pharmacologic interventions. The addition of a human preclinical model of compatible xenotransplantation to existing NHP model systems augments our knowledge of immune responses to porcine xenografts and will likely play a key role in the propulsion of experimental porcine xenotransplantation to clinical reality.

**Fig. 4 | Limited infiltration of the porcine kidney xenograft by human myeloid immune cells.** Spatial transcriptomics was performed on serial needle core biopsies of 10-GE porcine kidneys before and after transplantation into a brain-dead human recipient. Biopsies were obtained from either the right (pre-transplant and day 3 samples) or left (day 1 and day 3T) xenografts. Cell type signatures were identified from reference transcriptomes using cell2location. **a** Human myeloid cells are detected in biopsies of the porcine kidney xenograft three days after transplantation. Capture spot color corresponds to cell abundance, and color scales to the right of each spatial plot indicate cell abundance. Note that scaling is conserved across time for a given cell type but differs between macrophages (cell abundance range: 0–2) and neutrophils (cell abundance range: 0–1). Visualization of cell abundance in a given capture spot is thus capped at 1 or 2 cells. **b** Calculated total (*left*) and normalized (*right*) cell abundance for various immune cell types in the indicated biopsies. For clarity, quantification of B cells is not shown. **c** Tissue zones and spatial co-localization of immune and kidney parenchymal cells were determined using the non-negative matrix factorization (NMF) of the deconvolution output in cell2location. The dot plot represents relative NMF weights of detected cell types (rows) across NMF "facts" (NMF factor) that correspond to cellular compartments/zones. Co-localized cell types can be found within each respective NMF compartment/zone. Of note, the size of the day 1 biopsy (<200 capture spots) did not permit co-localization analysis. The dot color and size is a representation of the proportion of cells of each respective cell type. Pre-tx pretransplant. Day 3T biopsy was taken on post-transplant day 3 at study termination.

## Methods

### Porcine kidney xenotransplantation

Our research complies with all relevant ethical regulations. Pig-to-human kidney xenotransplantation in the preclinical human decedent model ("Parsons Model") was conducted in accordance with the University of Alabama at Birmingham (UAB) IRB-300004648. In brief, kidney procurement from a pig (Male Chester-White crossbreed, 368 days old, 159 kg) with 10 gene edits ("10-GE") was performed using aseptic technique in a surgical suite adjacent to a facility free of designated pathogens on the Xenotransplantation Procurement Campus (XPC) of the University of Alabama at Birmingham Heersink School of Medicine (UAB)[3]. Oversight of all activities at the UAB XPC was provided by the Institutional Animal Care and Use Committee (IACUC-22015).

Eligible human decedents were adults age 18 years or older who were declared brain-dead and referred for organ donation, but ruled out for donation of heart, lung, liver, pancreas, and/or intestine[3]. The recipient of the porcine kidney xenograft was a 57-year-old brain-dead human male who died of blunt trauma. After declaration of the brain death, the local organ procurement organization (Legacy of Hope) offered the decedent's organs for clinical transplantation. The decedent's family was approached to participate in this study after the liver and thoracic organ wait lists were exhausted and provided consent in accordance with CARE guidelines and the Declaration of Helsinki principles. The decedent's next-of-kin authorized research and transport to the Legacy of Hope donor recovery center at the University of Alabama at Birmingham[3]. After completion of a negative prospective flow cytometric crossmatch[3] with the donor pig, the recipient underwent native nephrectomies followed by transplantation of right and left 10-GE porcine kidney xenografts. Of note, the decedent's native kidneys were removed for the purposes of transplantation but were ultimately declined by all transplant centers. The native kidneys were thus utilized as controls in this study according to the wishes and consent of the decedent's family. The brain-dead human recipient was maintained in an operating room in the donor recovery center and supported by various intensive care interventions (e.g., ventilation, pharmacologic pressors, etc.) until termination 74 h after transplantation of the porcine kidney xenografts[3]. Pharmacologic immunosuppression consisted of induction therapy with methylprednisolone, anti-thymocyte globulin (ATG), and rituximab (anti-CD20), while maintenance immunosuppression included tacrolimus, mycophenolate mofetil and prednisone (Table 1 and ref. [10]). Additional methylprednisolone doses as well as phenylephrine, vasopressin, and levothyroxine were given for management of brain death[3].

### Sample collection

**Porcine kidney xenograft biopsies.** Core biopsies of the 10-GE porcine kidney were collected prior to transplantation and at 24, 72, and 74 h post transplantation using a core biopsy needle. All post-transplantation core biopsies were collected from the xenograft in situ in the brain-dead human recipient. In addition, a wedge biopsy was taken from the explanted porcine kidneys and flash frozen in liquid nitrogen. Core biopsy samples were placed into PBS on ice and transferred to the laboratory where they were immediately embedded in Optimal Cutting Temperature and flash frozen in a 2-methylbutane container surrounded by liquid nitrogen. Frozen samples were stored in blocks at −80 °C until sectioned. A section of each biopsy was placed on a Visium gene expression slide (10X Genomics, Cat. No. 1000187 or 1000094) for spatial transcriptomic analysis. The remainder of each biopsy was committed to nuclear isolation and single-nuclear RNA-seq as below.

**Porcine kidney xenograft explant processing and immune cell isolation.** A section of the explanted pig kidney was chosen to represent the kidney from the cortex through the papillary region. The kidney section was finely minced using a razor blade and placed into a medium containing 16.7 units per mL Liberase (Millipore Sigma, Cat. No. 5401119001) in RPMI 1640 (Gibco ThermoFisher Scientific, Cat. No. 11875135) for 30 min at 37 °C. The reaction was stopped by adding phosphate-buffered saline (PBS) (Gibco ThermoFisher Scientific, Cat. No. 10010049 with 1% w/v bovine serum albumin (BSA) (Fisher Scientific, Cat. No. BP9706100) and then tissue was pulled through an 18-gauge syringe to dissociate the remaining tissue to a single-cell suspension. Cells were pelleted at 500 × *g* for 5 min and resuspended in ACK lysis buffer (Quality Biological, Cat. No. 118-156-101) for 2 min to lyse red blood cells. Cells were washed with 45 mL of PBS and then stained with 1 μL per 1 × 10 cells[6] Aqua fixable viability dye (Thermo-Fisher Scientific, Cat. No. L23105). Cells were incubated in 5 μL per 100 μL cell suspension Human TruStain FcX, Fc Receptor blocking solution, (Biolegend Inc., Cat. No. 422302) for 10 minutes at room temperature and then stained with 5 μL per test of 0.5 mg per mL anti-human CD45 FITC clone 2D1 (Biolegend Inc., Cat. No. 368508) and 10 μL per 100 μL cell suspension of mouse anti-pig CD45-Alexa Fluor 647 conjugate antibody clone K252.1E4 (Bio-Rad Laboratories, Inc, Cat. No. MCA1222A647). Cells were sorted using a BD FACSAria for human and pig CD45+ cells into a tube containing PBS + 0.04% w/v BSA.

**Peripheral blood mononuclear cells (porcine & human).** Blood was collected in EDTA-K2 tubes vacutainers (BD and Company, Cat. No. 367863). PBMC isolation was performed at room temperature until red blood cell (RBC) lysis. Equal volumes of DPBS without Ca2+ or Mg2+ (Gibco ThermoFisher Scientific, Cat. No. 14190094) and 2% FBS (Gemini Bio, Cat. No. 100-106) was added to the whole blood and mixed (i.e. 5 mL of whole blood was diluted with 5 mL of buffer). The manufacturer's instructions were followed in order to isolate PBMCs using Lymphoprep (StemCell Technologies, Cat. No. 07851) and either a Sepmate-15 (StemCell Technologies, Cat. No, 85415) or a Sepmate-50 (StemCell Technologies, Cat. No, 85450) depending on the blood volume. Once the PBMCs were isolated, the cells were washed with buffer two times, once at 300 × *g* for 8 min and once at 120 × *g* for 10 min with no brake. After removing the wash buffer, RBCs were lysed using 4 mL of room temperature ACK lysis buffer (Quality Biological; Cat. No. 118-156-101) for 2 minutes on ice and then the ice cold DPBS was added to the 14 mL mark and mixed. Cells were pelleted by centrifugation, 400 × g for 5 min at 4 °C. The buffer was removed and

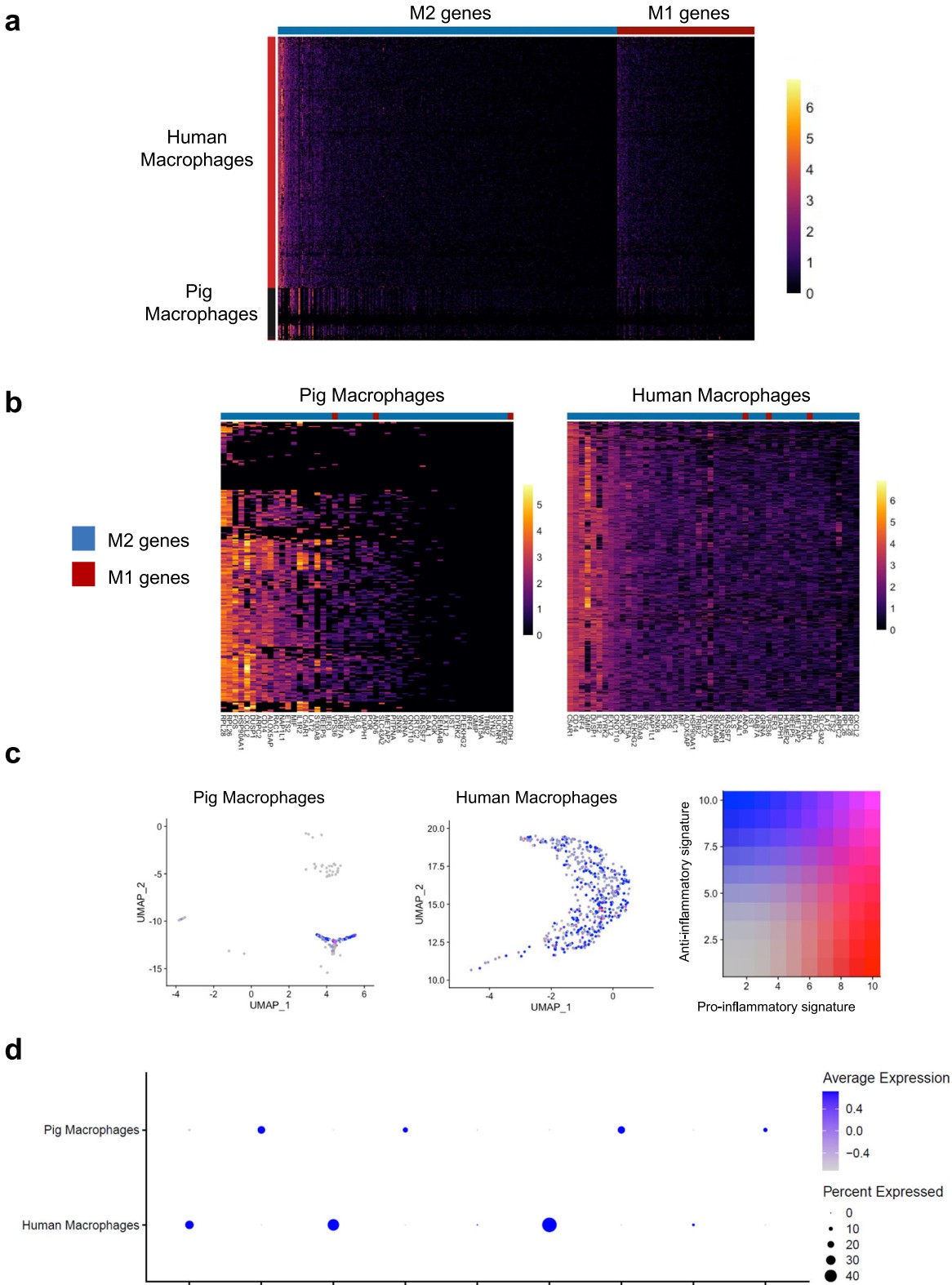

**Fig. 5 | Predominance of M2-like macrophages in the porcine kidney xenograft.** scRNA-seq was performed on CD45+ immune cells sorted from the right porcine kidney xenograft at explant (see Supplementary Fig. 3), and macrophage clusters were selected for analysis. **a** Expression of M1 (red) and M2 (blue) genes in human and pig macrophages (see Supplementary Dataset 1 and ref. 16 for full gene list). **b** Expanded view of top 50 most highly expressed M1 and M2 genes in each species. Note M2 > M1 genes for both species. **c** Composite gene expression score of pig and human macrophages of M1-like pro-inflammatory (red) and M2-like anti-inflammatory (blue) gene signatures[16]. UMAPs were generated from re-clustering of macrophage clusters selected from Supplementary Fig. 3. **d** Expression of select anti- and pro-inflammatory cytokine genes in pig and human macrophages. Average gene expression is visualized such that the mean of the scaled expression dataset is set at 0 with a standard deviation of 1. *ss11-IL6* was not detected.

0.5–1 mL fresh, ice cold DPBS without Ca2+ and Mg2+ was added and the cells were suspended. Cells were counted and delivered to the UAB Single Cell core, and scRNA-seq was performed.

**Human kidney.** The native kidneys which were removed from the brain-dead recipient prior to xenotransplantation were offered for allotransplantation but ultimately declined. After exhaustion of the transplant list, the kidneys were transported to the laboratory and processed for spatial transcriptomics analysis as described below.

**Control pig kidneys.** Porcine control kidneys were recovered from (1) a 208-day-old Chester White Cross wild-type sow (Identification number 817D), weighing 154 kg, and (2) an 8-day-old Chester White Cross 10-GE male piglet (identification number 817D-1) weighing 1.3 kg at the UAB XPC. The kidneys were transported to the laboratory and processed for spatial transcriptomics analysis as described below.

### Spatial transcriptomics – sample preparation

Frozen biopsy OCT blocks were equilibrated to −10 °C before use. Using a cryostat, a 10 µm section was placed onto a Visium Spatial Gene Expression Slide and processed according to the manufacturer's protocol. In summary, slides were fixed in methanol for 30 min and stained with Hematoxylin and Eosin. Brightfield images were taken using a BZ-X700 microscope (Keyence Corporation of America). Slides were placed in specialized slide cassette holders and permeabilization enzyme was added for 12 min at 37 °C to release the RNA onto the slide. Using the captured RNA, cDNA and subsequently second-strand DNA was created and amplified. Libraries were generated and sent for paired-end sequencing on the NovaSeq6000 (Illumina) for a depth of 50,000 reads per capture spot.

### Nuclear isolation for single-nuclear RNA-sequencing

After a section was taken for spatial transcriptomics, the remaining xenograft biopsy tissue was thawed from OCT and washed with PBS. Nuclei were subsequently isolated using Nuclei Lysis Buffer containing Nuclei Isolation Kit: Nuclei EZ Prep Buffer (Millipore Sigma, Cat. No. NUC101-1KT) supplemented with EDTA-free cOmplete ULTRA Tablets (Millipore Sigma, Cat. No. 05892791001) and Invitrogren SUPERase IN (ThermoFisher Scientific, Cat. No. AM2682) and RNAsin Plus Ribonuclease inhibitors (Promega, Cat. No. N2611). Tissue was minced into <1 mm pieces in 2 mL of Nuclei Lysis Buffer. Samples were transferred to a Kimble Dounce homogenizer (MilliporeSigma, Cat. No. D8938-1SET) and homogenized. An additional 2 mL of Nuclei Lysis Buffer was added to the sample and incubated for 5 min on ice. Samples were passed through a 40 µm pluriStrainer (pluriSelect Life Science, Cat. No. 43-50040-51) into a 50 mL conical tube. Samples were centrifuged at 500 × $g$ for 5 min at 4 °C. The supernatant was removed and the pellet was washed with 4 mL of Nuclei Lysis Buffer containing 1% BSA for 5 min on ice. Samples were centrifuged at 500 × $g$ for 5 min at 4 °C. Samples were passed through a 5 µm filter into a 50 mL conical tube and then centrifuged again. Nuclei were resuspended in a solution containing PBS, 1% BSA, and 0.1% RNAse inhibitor.

### Preparation of gel emulsion microdroplets and single-cell/single-nuclear RNA sequencing

Suspensions of single cells (or nuclei) were placed on ice and transferred to the UAB Flow Cytometry and Single Cell Core where they were immediately processed using a Chromium 3′ Single-Cell RNA reagent kit v3 (10X Genomics, Cat. No. 1000268) according to the manufacturer's protocol. In summary, cells or nuclei were counted and loaded onto the Chromium Controller (10X Genomics), and gel emulsion microdroplets were prepared. cDNA was generated and amplified from mRNA collected within each microdroplet. Libraries were generated and sent for paired-end sequencing on a Nova-Seq6000 (Illumina) for a depth of 20,000 reads per cell.

### Generation of the hybrid human-pig genome reference

The human GRCh38 (GCA_000001405.28) and porcine *Sus scrofa* 11.1 (GCA_000003025.6) fasta and gene annotation (gtf) files were downloaded from Ensembl. Gtf files were filtered according to the standard 10X Genomics protocol (https://support.10xgenomics.com/single-cell-gene-expression/software/pipelines/latest/advanced/references), including protein-coding, lincRNA, antisense, and immunoglobulin genes. The two genomes were merged using the Cell Ranger version 6.0 *mkref* function to create the hybrid hg38-ss11 genome.

### Analyses using the hybrid human-pig reference genome
**Datasets.**
- scRNA-seq library prepared from CD45+ immune cells sorted from the explanted xenograft (Figs. 1–5, Supplementary Figs. 4–7);
- snRNA-seq libraries prepared from single nuclei isolated from core and wedge biopsies of the porcine kidney xenograft (Supplementary Figs. 7 and 8);
- Spatially barcoded libraries prepared from core biopsies of the porcine kidney xenograft (Figs. 3 and 4; Supplementary Figs. 7 and 8).
- Spatially barcoded libraries prepared from control kidney samples, including human kidney, 10-GE pig kidney, and wild-type pig kidney (Supplementary Fig. 3).

**scRNA-seq data processing and analysis of CD45+ cells sorted from the porcine xenograft at explant.** Cell Ranger (v.6.1.1) was used to pre-process sequenced reads aligned against the hybrid human-pig genome reference to generate raw count matrices. Downstream analysis on the filtered UMI expression profile for each droplet was completed using R (v.4.2.1) and Seurat[33,34] (v.4.2.0) using default parameters unless otherwise specified. Before conducting additional analyses, background noise from ambient RNA was removed using SoupX[35] (v.1.6.1). The overall contamination fraction (rho) was parameterized using the *autoEstCont* function to remove > 2% background contamination in our dataset. The SoupX-corrected count matrix was then loaded into R using the *Read10X* function and was further used to create a Seurat object. Cells with <200 unique features, > 3000 unique features, and > 12% mitochondrial gene expression were filtered. Features expressed in <5 cells, and cells with doublet scores >0.3 were also removed. Data were then normalized by a scale factor of 10,000 and log1p-transformed using the *LogNormalize* function. We then identified the top 3000 variable genes, ranked by coefficient of variation, using *FindVariableFeatures*. Using the variable genes, we scaled and centered the genes across the cells using the *ScaleData* function followed by the identification of principal components (*RunPCA*). Thirty principal components were found and used to construct a k-nearest neighbor (KNN) graph using *FindNeighbors*. Clustering was subsequently performed using *FindClusters* which employs a shared nearest neighbor (SNN) modularity optimization based clustering algorithm. Cluster information was used as input into the uniform manifold approximation and projection algorithm (*RunUMAP*) which further aided the visualization of cell manifolds in a low-dimensional space. We ran Seurat's implementation of the Wilcoxon rank-sum test (*FindMarkers*) to identify differentially expressed genes in each cluster. The expression of cluster-specific canonical markers was used to annotate each cell cluster.

Module scores were determined for the Seurat object using the *AddModuleScore* function with the M1/pro-inflammatory or M2/anti-inflammatory gene lists as inputs (Supplementary Dataset 2 and ref. 26). A feature plot was generated using *FeaturePlot* with the features set as the calculated module scores from each gene list.

**snRNA-seq data processing and analysis of parenchymal cells from needle core biopsies of the porcine kidney xenograft.** Sequenced reads from nuclei recovered from the four needle core and core biopsies of the porcine xenograft in situ (see above) were processed

using Cell Ranger (v.6.1.1) and aligned to the hybrid human-pig reference genome. Generated snRNA-seq counts from each sample were further processed using Seurat (v.3.2.3) and its associated dependencies. Specifically, filtered UMI counts from each time point were imported into R using the *Read10X* function and structured into sample-specific Seurat objects (*CreateSeuratObject*). Generated objects from each time point were labeled with unique group identifiers and merged into a single object using Seurat's *merge()* function while including genes detected in at least 5 cells. The merged dataset was further filtered to retain cells with unique feature counts over 200 or under 2500, and cells with a fraction of nuclear-encoded mitochondrial genes <15%. Data were normalized by a scale factor of 10,000 and log1p-transformed using *LogNormalize()* before identifying variable genes. Data were scaled, centered, and principal components were identified as detailed above. Data were then integrated using Harmony[36] (v.0.1.0) followed by Nearest Neighbor analysis (*Find-Neighbors*) and dimensional reduction using uniform manifold approximation and projection (*RunUMAP*). Cell clusters were identified using *FindClusters*. Differential expression (DE) analysis between clusters/two groups of cells was performed using the non-parametric Wilcoxon Rank Sum test implemented in the FindMarkers() functions from the Seurat R package. DE testing was limited to genes which show, on average, at least 0.1 log-fold change between the two groups of cells and on genes that are detected in a minimum fraction of 0.01 cells in either of the two populations. A gene was defined as differentially expressed if the absolute average log fold-change (avg_logFC) was >0.25 and the Bonferroni-adjusted *p*-value < 0.01.

### Visium spatial gene expression data processing and analysis using Cell2location

**Cell-type deconvolution of spatial transcriptomics data using Cell2location.** To spatially resolve cell populations identified in the kidney, pre- and post-transplant, we used cell2location[20], a Bayesian model which estimates cell type abundance by deconvoluting a spatial expression count matrix into a set of reference cell type signatures. The model takes a spatial expression matrix with mRNA counts of genes at spatial locations and a matrix of reference cell type signatures as input. All cell2location analyses were conducted using Scanpy[37] (v. 1.9).

**Construction of snRNA-seq and scRNA-seq reference transcriptomes.** Reference transcriptomes input into Cell2location derived from the sorted CD45+ cells (scRNA-seq) (*n* = 6513) and the nuclei isolated from the core and wedge biopsies of the porcine kidney taken pre-transplant and on Day1, day 3 and day 3T post transplant (*n* = 7868). These data were pre-processed as described above for scRNA-seq. Cells from both datasets were merged into a unified object using Seurat's *merge* function yielding a total of 14,381 cells and 27,536 genes in the reference dataset. The merged dataset was then normalized, scaled, and centered as described above. Thirty principal components were found and used to construct a k-nearest neighbor (KNN) graph and clustering was performed using modularity optimization, resulting in 17 reference cell clusters which were ultimately collapsed to broader cell types (*n* = 13 clusters). Cell clusters were visualized in low-dimensional space on a UMAP, and marker genes were used to annotate each cell cluster in the reference dataset (Supplementary Fig. 7).

**Estimating reference cell type signatures.** *The snRNA-seq and scRNA-seq reference data* and corresponding reference cell type annotations were loaded into Scanpy using *sc.read_10x_mtx*. Gene selection and general QC were conducted as described before. Expression signatures of the 13 reference cell types in the reference dataset were then estimated using the negative binomial regression model while accounting for batch effects. The created model was further trained to estimate the reference cell type signatures on all cells in the dataset

(train_size = 1), mini-batch size = 2500 and a training duration of 600 epochs, with GPU acceleration. Model accuracy was evaluated based on the evidence lower bound (ELBO) loss and reconstruction accuracy plots.

**Spatial mapping of kidney reference cell-types to spatially barcoded xenotransplant biopsies using cell2location.** We imported the Visium spatial transcriptomics data for each biopsy from 10X Space Ranger into Scanpy, using *scanpy.read_visium*. Mitochondrial genes were identified and removed. Before running cell2location to map our reference cell types to each Visium dataset, both the estimated reference cell type signatures (gene expression), and spatial data from each respective biopsy were filtered to select genes shared between the two data objects. The filtered objects were both used as input to cell2location.

We specified two hyperparameters required to run cell2location: (1) The expected cell abundance per location (N_cells_per_location), which was estimated by manually counting nuclei in 10 random capture spots from the respective H&E images and averaging these values to give us the average cell abundance per tissue sample (Pre-transplant = 6.6 cells/capture spot; day 1 = 5.7 cells/capture spot; day 3 = 5.6 cells/capture spot; day 3T = 5.4 cells/capture spot); and 2) a parameter which accounted for within-experiment variation in RNA detection sensitivity (*detection_alpha*). We set detection alpha = 20 in order to ensure the greatest accuracy and sensitivity given that strong gradients in mRNA detection sensitivity are commonly observed in adult human 10x Visium data.

After providing the reference cell type signatures, spatial data, and hyperparameters specified above as inputs, the cell2location model was trained using max_epochs = 30000 on the full dataset (batch_size = None) while estimating cell abundance at all locations (train_size = 1). We plotted ELBO loss history during training and assessed mapping quality by examining reconstruction accuracy plots. We conducted four runs using the same reference dataset for each biopsy taken at the four different time points (Pre-transplant, day 1, day 3 and day 3T). We exported estimated posterior distributions of cell abundance ('num_samples': 1000, 'batch_size': mod.adata.n_obs, 'use_gpu': True) and added a 5% quantile of the posterior distribution, representing the value of cell abundance corresponding to high confidence in the model. We further leveraged Scanpy's plotting function *scanpy.pl.spatial* to visualize spatial scatter plots of cell type abundance in spatial coordinates.

**Inference of spatially co-located cell types using non-negative matrix factorization (NMF).** Tissue zones and spatial co-localization of immune and kidney parenchymal cells were determined using non-negative matrix factorization (NMF) from the scikit-learn package and implemented within cell2location. Cell type density/abundance estimates from cell2location were used as input for NMF to identify groups of co-located cell types (tissue zones/cellular compartments). To find strong co-location signals, and to select the most meaningful cell type groups per biopsy, NMF was trained on a range of factors R = {5,...,30}; number of training restarts = 3 (otherwise, default model parameters were used). The density of each cell type across tissue locations was modeled as an additive function of the cell. The day 1 biopsy contained too few capture spots to perform co-localization analysis. Based on the cell2location outputs that show no presence of human immune cells in the pre-transplant biopsies, the NMF co-localization was performed using only porcine cells for the pre-transplant biopsy.

### Spatially barcoded libraries prepared from control kidney samples, including human kidney, 10-GE pig kidney, and wild-type pig kidney

**Data pre-processing.** Base call (BCL) files were converted to FASTQ files using the Space Ranger version 1.3 *mkfastq* function. Space Ranger

*count* function was used to align the FASTQ files to the hybrid hg38-ss11 genome to generate count matrices.

**Data analysis.** Spatial transcriptomics analyses were carried out using packages created for the R statistical analysis environment (v. 4.2.1). Data were primarily analyzed using Seurat (v. 4.2.0) and its associated dependencies. Data from each sample were imported and structured into a Seurat object using the *Load10X_Spatial*. Data were further filtered (nCount_Spatial > 1) and normalized using *SCTransform* before visualizing gene expression in the respective spatial transcriptomics landscapes.

### Species assignment validation analyses using the modified species-specific reference genome
#### Datasets.
- scRNA-seq libraries prepared from pre-transplant porcine and human PBMCs
- scRNA-seq library prepared from CD45+ immune cells sorted from the explanted xenografts

**Construction of the species-specific reference genomes and data analysis.** Sequenced reads were processed with Cell Ranger (version 6.1.1). Reference genomes utilized included the hg38 and ss11 genomes as provided by Ensembl as well as the pre-compiled hg38 reference genome provided by 10x Genomics. Pre-transplantation porcine and human samples were initially mapped to the opposite species' reference genome to identify genes. Any gene with >3 counts assigned was then subsequently identified and removed from the original reference, thereby resulting in a modified reference genome for each species. All samples were subsequently mapped to both species' modified reference genomes and gene mapping rates on a per cell basis were compared between the two to enable identification of porcine cells from human. As some level of mapping consistently occurred when processing samples against the opposite species' reference genome, cells were identified as porcine if they had a ratio of human to porcine mapped genes <0.75 and subsequently identified as human if the ratio was >1.33. Cells that fell between the two ratios were labeled as ambiguous and excluded from analysis. Samples were then mapped to the original unmodified reference genomes of each species. For post-transplantation human blood samples, cells identified as porcine from the modified reference mapping, were filtered out from the unmodified human reference mapped data and included from the unmodified porcine reference mapped data for final analysis. Downstream analysis was completed using R (v.4.2.0) and Seurat (v.4.1.1) using default parameters unless otherwise noted. Samples were normalized and scaled after standard QC measures including filtering of cells with less than 200 features, removing features with less than 3 cells expressing them, removing immunoglobulin genes and removing cells that had over 8% mitochondrial gene expression. Samples were integrated using Harmony (v.0.1).

### BLAST analysis
Nucleotide basic local alignment search tool algorithm blastn (https://blast.ncbi.nlm.nih.gov) was used to generate pairwise alignments of human and porcine sequences.

### Reporting summary
Further information on research design is available in the Nature Portfolio Reporting Summary linked to this article.

## Data availability
The human and wild-type control pig kidney data generated in this study have been deposited in the Gene Expression Omnibus database under accession code GSE242270. The data from the 10-GE pig have been deposited to the European Genome-Phenome Archive (EGA) Database under study accession code EGAS50000000244 and data accession code EGAD50000000359 and are available under restricted access due to privacy restrictions given the proprietary nature of the 10-gene edited porcine kidney product. Data access can be obtained by reaching out via email to the corresponding author (pmporrett@uabmc.edu) who will provide instructions within one week to the requestor so that the requestor may apply for data access from the EGA. In brief, the requestor will need to review and agree to the conditions of the EGA Data Access Agreement. All other data are available in the article and its Supplementary files or from the corresponding author upon request. Source data are provided with this paper.

## Code availability
All analyses were performed using open-source packages, and details of commands used in the packages are provided in the Methods. All analysis code is available on Github at: https://github.com/PorrettLab/Spatiotemporal-immune-atlas-of-the-1st-clinical-grade-gene-edited-pig-to-human-kidney-xenotransplant/tree/main[38].

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

## Acknowledgements

The authors are deeply indebted to the donor family of for honoring the donor's wishes to become an organ donor for the purposes of transplantation and/or research. The authors would also like to thank dozens of critical partners at the Legacy of Hope organ procurement center; the UAB operating room nursing staff; staff members at the UAB Xenotransplantation Procurement Campus; and the UAB Division of Transplantation surgical faculty. We would additionally like to thank Drs. Anupam Agarwal (UAB), Bradley Yoder (UAB), and David Ayares (Revivicor, Inc.; Blacksburg, VA) for provision of reagents, equipment, and animals; Dr. Caroline Kelly for critical reading of the manuscript; and Connor Strickland for computational assistance. Finally, the authors would like to extend special thanks to members of the 10X Genomics team (Pleasanton, CA) for technical and analytic expertise as well as logistic support, including Sai Saroja Kolluru, Nirav B. Patel, Andrew W. Greer, Dr. Soo Hee Lee, and Dr. Ryan Mote. Supplemental Figs. 1a and 7a were created with biorender.com. This work was sponsored by a grant from United Therapeutics. R.A., E.D.W., C.F.F., D.E., B.J.O., M.B., G.B., J.P., R.R., S.C.L., A.F.R., J.E.L., and P.M.P. all received salary support from United Therapeutics. The authors would like to acknowledge the following sources of salary support: T32-AI007051 and F30-DK132814 to MDC; T32-GM-008361 to M.D.C., G.G., and M.E.G.; American Heart Association Predoctoral Fellowship 827257 to E.N.E.; and T32 DK007545 to K.H.M.

## Author contributions

M.D.C. and R.A. contributed equally to this study. M.D.C., R.A., J.E.L. and P.M.P. conceived and designed the study. M.D.C., R.A., E.N.E., C.F.F., S.L., C.S., V.S.H., H.C.P., E.D.W., G.G., D.E., B.J.O., V.K., D.J.A., M.E.G., M.B., S.Y., K.H.M., J.L., J.T.K., G.B., J.P., Z.K., R.R., S.C.L., A.F.R., J.F.G., J.E.L. and P.M.P. were involved in data collection and management. M.D.C., R.A., C.F.F., S.C.L., A.F.R. and P.M.P. performed data analysis. M.D.C., R.A., C.F.F., D.E., S.C.L. and P.M.P. wrote the manuscript. M.D.C., R.A., E.N.E., C.F.F., S.L., C.S., V.S.H., H.C.P., E.D.W., G.G., .DE., B.J.O., V.K., D.J.A., M.E.G., M.B., S.Y., K.H.M., J.L., J.T.K., G.B., J.P., Z.K., R.R., S.C.L., A.F.R., J.F.G., J.E.L. and PMP discussed the results and edited the manuscript.

## Competing interests

The following authors receive or have received salary support from a research grant from United Therapeutics: R.A., E.D.W., C.F.F., D.E., B.J.O., M.B., G.B., J.P., R.R., S.C.L., A.F.R., J.E.L., and P.M.P. The remaining authors declare no competing interests.
