## [Peer Review File · Nature Communications]

Spatiotemporal immune atlas of a clinical-grade gene-edited pig-to-human kidney xenotransplantEditorial Note: This manuscript has been previously reviewed at another journal that is not operating a transparent peer review scheme. This document only contains reviewer comments and rebuttal letters for versions considered at *Nature Communications*.

REVIEWER COMMENTS

Reviewer #3 (Remarks to the Author):

In the revision, the authors provide QC metrics which show that the day 1 sample is of low quality and the other samples are of medium quality by current standards.

It does not logically follow that because the pig kidney is a commercial product that the transcriptomic data cannot be publicly released.

Regarding analysis of the pig parenchyma in the Visium samples and attempting to colocalize various detected immune cell types with various pig nephron segments whether injured or not, the authors defer on the grounds that (1) this will be shown in a different manuscript, (2) common histology would better show the relationship between immune cells and parenchyma. Regarding (1), the authors already show pig parenchyma Visium data in Extended Fig 8 – it is already part of this paper. Further, the title states that this is a “spatiotemporal immune atlas” but without showing immune cells in context with parenchyma, there is no context to the spatial immune map, which makes it uninformative spatially. While low resolution, there are several existing informatic tools to assess enrichment of various immune cells vs. parenchymal cell states in Visium spots. Regarding (2), if histology would better show the response of the parenchyma to immune infiltration than Visium, then why perform Visium at all – what does it add to the paper? The authors could have simply performed histology to show that there is no evidence of parenchymal rejection.

Reviewer #4 (Remarks to the Author):

This is a report on the so-called decedent model: what was the reason for the brain death?

The results of model are not FDA relevant (IXA meeting, San Diego 2023). The descendant got only cortisone treatment. No T3,,T4, vasopressin. Organ donation was "ruled out". Why were the decent's kidneys still offered?

Introduction too long and not fair: there are consistent heart and (Yamada) kidney results using co-stimulation blockade.

Pig kidney and recipient were compatible? What kind of tests?

IS, MP is missing the description of the induction.

TM (and tubular necrosis) shouldn't happen in 10GM donor. Poss reason cloning technique.

My conclusion: TKO and two complement regulators prevent hyperacute and delayed acute humoral rejection reactions. No human B cells. 72 h too early for a cellular T cell reaction. So, their conclusion, co-stimulation blockade not necessary is brave.

Nice and unique basic research piece. Diff. porcine vs human

REVIEWER COMMENTS

Please note our responses in red.

Page & line numbers below refer to the “tracked” manuscript document.

Reviewer #3 (Remarks to the Author):

In the revision, the authors provide QC metrics which show that the day 1 sample is of low quality and the other samples are of medium quality by current standards.

It does not logically follow that because the pig kidney is a commercial product that the transcriptomic data cannot be publicly released.

Regarding analysis of the pig parenchyma in the Visium samples and attempting to colocalize various detected immune cell types with various pig nephron segments whether injured or not, the authors defer on the grounds that (1) this will be shown in a different manuscript, (2) common histology would better show the relationship between immune cells and parenchyma. Regarding (1), the authors already show pig parenchyma Visium data in Extended Fig 8 – it is already part of this paper. Further, the title states that this is a “spatiotemporal immune atlas” but without showing immune cells in context with parenchyma, there is no context to the spatial immune map, which makes it uninformative spatially. While low resolution, there are several existing informatic tools to assess enrichment of various immune cells vs. parenchymal cell states in Visium spots. Regarding (2), if histology would better show the response of the parenchyma to immune infiltration than Visium, then why perform Visium at all – what does it add to the paper? The authors could have simply performed histology to show that there is no evidence of parenchymal rejection.

We apologize for any confusion caused by our previous response. When referring to “microscopy studies”, we were referring to immunofluorescence microscopy (IF) as a method to examine immune cell architecture within various kidney segments. We believe this method would be ideal for identifying spatial interactions of immune cell types with specific kidney parenchymal cells. However, identification of species-specific cell types with IF is not currently possible given the reagent limitations which we outlined in our previous response. Thus, single-cell and spatial transcriptomic approaches are best suited at the present time for these analyses.

We greatly appreciated the reviewer’s suggestion that we could enhance the spatial information of this work by providing context to the spatial immune map. To this end, we used cell2location to identify cellular compartments and co-occurrence of cell types in the biopsy sections. We then used a non-negative matrix factorization (NMF) approach (similar to SPOTlight) to determine that the majority of human immune cells co-localized with porcine endothelial and stromal cells. Meanwhile, most of the porcine immune cells were found with other kidney parenchymal cells in the loop of Henle and proximal tubules. These new results can be found in Figure 4C and in the accompanying manuscript text (page 9, line 227-234). We have further updated the Methods (pages 27-28, line 637-648) and provided additional commentary in the Discussion (page 12, lines 303-310).

Reviewer #4 (Remarks to the Author):

This is a report on the so-called decedent model: what was the reason for the brain death? The results of model are not FDA relevant (IXA meeting, San Diego 2023). The descendant got only cortisone treatment. No T3, T4, vasopressin. Organ donation was “ruled out”. Why were the decedent’s kidneys still offered?

We thank the reviewer for these comments.

Brain death was due to blunt head trauma. We provided this detail in our original publication (Porrett et al., *Amer J Transpl*, 2022), which we have cited in this manuscript (page 16, lines 385-397) and added to the Methods for this work (page 16, lines 387-388). Of note, the decedent was indeed maintained on phenylephrine, vasopressin, levothyroxine, and methylprednisolone as per routine management of brain-death before organ procurement. This detail has been added to the manuscript as well in the Methods section (page 17, lines 404-405). We had previously described the potential impact of brain death on our study’s findings in the discussion and have preserved these comments in the Discussion (page 14, line 354-365).

We appreciate the opportunity to clarify the process of organ donation and how this decedent was selected for this study. In brief, the organ procurement organization (OPO) approached the decedent’s family about potential organ donation for the purposes of clinical transplantation or research after notification of his demise. Per their protocols, the OPO offered the decedent’s organs for clinical transplantation; however, all organs were ultimately declined for transplantation by all transplant centers. We refer to this process in clinical transplantation as exhaustion of the wait list. Additional details pertaining to this case have now been included in the Methods for clarity (page 16, lines 387-398).

Introduction too long and not fair: there are consistent heart and (Yamada) kidney results using co-stimulation blockade.

We thank the reviewer for this perspective and have shortened the introduction accordingly (page 4, lines 89-99). In the interests of brevity, we regret that we cannot detail all the critical contributions made by prior investigators to the field in our manuscript. Instead, we elected to cite a recent review² and a meta-analysis³ that incorporates the important work of Dr. Yamada and others. We meant no disrespect to the field as we outlined the rationale for work in human systems. To this point, we direct the reviewer’s attention to the opening sentence of the second paragraph of our introduction, where we maintain in this revision our prior emphasis on the critical contributions of NHP models (page 3, lines 69-71) to the field of xenotransplantation.

Pig kidney and recipient were compatible? What kind of tests?

We recently reported the development of a novel, prospective crossmatch flow cytometry assay¹. In summary, 10GE pig donor lymphocytes were combined with pre-xenotransplant

human decedent serum then incubated with FITC-conjugated goat anti-human IgG and then analyzed on the flow cytometer. Using this assay, we were able to perform a prospective crossmatch demonstrating lack of antibody binding in this decedent's serum to porcine cells. The results of this crossmatch assay, as well as the positive and negative controls, are detailed in Figure 2 of our clinical report¹.

IS, MP is missing the description of the induction.

If we have interpreted the reviewer's comment correctly, it is that methylprednisolone ("MP") is missing from the description of the immunosuppression ("IS") induction regimen in the Results. The reviewer is correct, and we apologize for our oversight in the text. We have thus added this to the Results (page 5, lines 126-128). We have not altered our Methods, where this agent was already included (page 16-17, lines 401-404).

TM (and tubular necrosis) shouldn't happen in 10GM donor. Poss reason cloning technique.

We presume that the reviewer's abbreviation of "TM" stands for thrombotic microangiopathy. As this experiment had not been performed in a human before, it was not clear what would happen histologically following xenotransplantation. As we stated in our original publication¹, we do not yet fully understand the etiology of the observed thrombotic microangiopathy. We agree with the reviewer that this is unexpected in the setting of these specific gene edits as well as crossmatch compatible transplantation. Acute tubular necrosis, however, can certainly be observed in allotransplantation even in the setting of short cold ischemic times. We agree with the reviewer that more data are needed to better understand the mechanistic basis of these observations, especially around the 10-GE kidney xenograft response to ischemia/reperfusion injury.

My conclusion: TKO and two complement regulators prevent hyperacute and delayed acute humoral rejection reactions. No human B cells. 72 h too early for a cellular T cell reaction. So, their conclusion, co-stimulation blockade not necessary is brave.

We agree that the experimental timeline may be too early for cellular infiltration and have previously acknowledged this in the Discussion (pages 11-12, lines 280-293, and page 13, lines 336-338). We recognize that this immunosuppression regimen is not in alignment with the traditional regimens used for NHP studies; however, we do not claim that co-stimulation is not necessary nor was our study designed to directly compare these regimens. We have added a comment in the Discussion to remind readers that our study does not rule out the use of co-stimulation blockade in future studies (page 14, lines 342-344).

Nice and unique basic research piece. Diff. porcine vs human

We greatly appreciate the reviewer's time and helpful feedback to improve the quality of our manuscript.

References:

1. Porrett, P. M. et al. First clinical-grade porcine kidney xenotransplant using a human decedent model. *Am J Transplant* 22, 1037–1053 (2022).

2. Sykes M, Sachs DH. Progress in xenotransplantation: overcoming immune barriers. *Nat Rev Nephrol.* 2022 Dec;18(12):745-761. doi: 10.1038/s41581-022-00624-6. Epub 2022 Oct 5. PMID: 36198911; PMCID: PMC9671854.
3. Firl DJ, Markmann JF. Measuring success in pig to non-human-primate renal xenotransplantation: Systematic review and comparative outcomes analysis of 1051 life-sustaining NHP renal allo- and xeno-transplants. *Am J Transplant.* 2022 Jun;22(6):1527-1536. doi: 10.1111/ajt.16994. Epub 2022 Feb 28. PMID: 35143091.

REVIEWERS' COMMENTS

Reviewer #3 (Remarks to the Author):

I have no further concerns.

Reviewer #4 (Remarks to the Author):

All my questions are answered satisfactorily

Reviewer #4 (Remarks on code availability):

Appropriate